

# Hypertension in frail older adults: current perspectives

Liying Li[1,*], Linjia Duan[1,*], Ying Xu[1], Haiyan Ruan[1,2], Muxin Zhang[1,3], Yi Zheng[1] and Sen He[1]

[1] Department of Cardiology, West China Hospital of Sichuan University, Chengdu, China
[2] Department of Cardiology, Traditional Chinese Medicine Hospital of Shuangliu District, Chengdu, China
[3] Department of Cardiology, First People's Hospital, Longquanyi District, Chengdu, China
* These authors contributed equally to this work.

Corresponding authors
Yi Zheng, drzhengyi1979@163.com
Sen He, hesensubmit@163.com

## ABSTRACT

Hypertension is one of the most common chronic diseases in older people, and the prevalence is on the rise as the global population ages. Hypertension is closely associated with many adverse health outcomes, including cardiovascular disease, chronic kidney disease and mortality, which poses a substantial threat to global public health. Reasonable blood pressure (BP) management is very important for reducing the occurrence of adverse events. Frailty is an age-related geriatric syndrome, characterized by decreased physiological reserves of multiple organs and systems and increased sensitivity to stressors, which increases the risk of falls, hospitalization, fractures, and mortality in older people. With the aging of the global population and the important impact of frailty on clinical practice, frailty has attracted increasing attention in recent years. In older people, frailty and hypertension often coexist. Frailty has a negative impact on BP management and the prognosis of older hypertensive patients, while hypertension may increase the risk of frailty in older people. However, the causal relationship between frailty and hypertension remains unclear, and there is a paucity of research regarding the efficacious management of hypertension in frail elderly patients. The management of hypertension in frail elderly patients still faces significant challenges. The benefits of treatment, the optimal BP target, and the choice of antihypertensive drugs for older hypertensive patients with frailty remain subjects of ongoing debate. This review provides a brief overview of hypertension in frail older adults, especially for the management of BP in this population, which may help in offering valuable ideas for future research in this field.

## INTRODUCTION

Hypertension is a prevalent chronic disease among the elderly, serving as a primary risk factor for cardiovascular disease, cerebrovascular disease, chronic kidney disease, and mortality in older people (*Mills, Stefanescu & He, 2020*; *Lawes, Vander Hoorn & Rodgers, 2008*). The prevalence of hypertension in people aged >60 years is estimated to exceed 60% (*Williams et al., 2018*), and it will continue to increase with the aging of the population

(*Mills, Stefanescu & He, 2020*; *Williams et al., 2018*). The global number of deaths attributed to hypertension and its associated complications exceeds 10 million every year (*Murray et al., 2020*). Reasonable management of blood pressure (BP) in older people is crucial for improving patient prognosis and alleviating the burden of healthcare system. Noticeably, elderly individuals are a unique population, and the management of BP is different from that of the general population. In addition, there is substantial heterogeneity in the biological age of older people due to chronic diseases and functional impairment, and chronological age does not accurately reflect their biological age (*Muller et al., 2014*).

Frailty is closely related to biological age (*Mitnitski et al., 2015*). It is an age-related geriatric syndrome that differs from disability and illness. The characteristics of frailty are decreased reserves of multiple organs and systems and increased sensitivity to stressors (*Dent et al., 2019*; *Clegg et al., 2013*), leading to the risk of adverse outcomes increased in older people, including mortality, dependence, hospitalization, and falls (*Dent et al., 2019*; *Clegg et al., 2013*; *Kojima et al., 2016*; *Hu et al., 2021*). Based on different definitions and assessment instruments of frailty, the prevalence of frailty ranged from 4.0% to 59.1% among community-dwelling older people (*Collard et al., 2012*; *Santos-Eggimann et al., 2009*; *He et al., 2019*). Frailty becomes progressively more common with advancing age: 4% for individuals aged 65–69 years, 7% for those aged 70–74 years, 9% for those aged 75–79 years, 16% for those aged 80–84 years, and 26% for those aged >85 years (*Clegg et al., 2013*).

Frailty and hypertension often coexist in older people (*Zhu et al., 2020*; *Vetrano et al., 2018*). As frailty is closely related to biological age, it can objectively reflect the health status and medical and healthcare needs of older people. Frailty has an important influence on the risk-benefit ratio of hypertension treatment in older people. However, there is a paucity of available research on the management of BP for frail older people. The benefits of treatment, the optimal BP target, and the choice of antihypertensive drugs for older hypertensive patients with frailty remain subjects of ongoing debate. Based on current evidence and guidelines, this narrative review provides a brief overview of hypertension in frail older people, especially for the management of BP for this population, which may help in providing valuable ideas for future research in this field.

## SURVEY/SEARCH METHODOLOGY

The authors conducted an in-depth search on PubMed, Web of Science, Embase, and the Cochrane Library. The search was carried out by combining subject words and free words, and the following heading terms were used when performing the search: "frail", "frailty", "old people", "old adults", "frail elderly", "hypertension", "hypertensive", "blood pressure", and "management of hypertension". The titles of the literatures underwent an initial screening, followed by a secondary screening of abstracts and keywords, and finally, the full texts were obtained for further evaluation. Additionally, relevant literature references were also searched to identify more eligible studies.

## THE ASSOCIATION BETWEEN FRAILTY AND HYPERTENSION

A systematic review revealed that the overall prevalence of frailty among hypertensive patients was 14.0%, and the overall prevalence of hypertension among frail patients was 72.0% (*Vetrano et al., 2018*). Frailty and hypertension share some common pathophysiological mechanisms, and hypertension and frailty are significantly correlated and influence each other in elderly people. However, the causal relationship between them remains unclear. On the one hand, frailty is associated with an increased risk of hypertension in older adults. For instance, a study conducted in Korea revealed that frail older adults had a significantly higher prevalence of hypertension than non-frail older adults (67.8% *vs.* 49.8%) (*Kang et al., 2017*). Another study from Brazil also showed that frail patients had a higher prevalence of hypertension than healthy patients (83.0% *vs.* 51.7%) (*Aprahamian et al., 2018*). Interestingly, *Aliberti et al. (2021)* found frailty could modify the association of hypertension with cognitive performance and impairment in older people, it showed that hypertension was associated with better memory and global cognitive scores among frail older adults. On the other hand, hypertension also increases the risk of frailty in older adults. For instance, a cohort study from China showed that hypertensive patients exhibited an approximately twofold greater prevalence of frailty than those non-hypertensive patients (13.8% *vs.* 7.4%) (*Ma et al., 2020*). In addition, there are some complex relationships between frailty and BP in older adults. The relationship between systolic BP (SBP) and frailty in hypertensive patients receiving antihypertensive therapy was found to exhibit a U-shaped pattern, as demonstrated by a study from Canada (*Rockwood & Howlett, 2011*). *Blauth et al. (2022)* performed a cross-sectional study in patients aged >80 years, and reported that frail patients had greater SBP during sleep than non-frail patients (128 ± 15 mm Hg *vs.* 122 ± 13 mm Hg). This study emphasizes the importance of sleep in investigations of frailty and hypertension, however, the sample size of this study was relatively small (38 frail and 36 non-frail patients) (*Blauth et al., 2022*), and more research can be conducted in the future. *O'Connell et al. (2015)* showed that frailty is related to a high risk of orthostatic hypotension.

Frailty has a significant impact on the prognosis of older hypertensive patients. Frailty markedly increases the risk of fall injuries (*Bromfield et al., 2017*), hospitalization (*Pajewski et al., 2016*), and mortality (*Li et al., 2023*; *Ma et al., 2018*) in elderly patients with hypertension, which seriously affects the quality of life of older people, reduces their independence, and decreases their life expectancy. Table 1 summarizes current research examining the impact of frailty on the prognosis of elderly patients with hypertension.

## FRAILTY ASSESSMENT IN OLDER HYPERTENSIVE PATIENTS

Frailty is affected by genetics, lifestyle, chronic diseases, environment and other factors (*Clegg et al., 2013*). It is closely related to biological age, and can objectively reflect the health status of older people. Assessing frailty status in older people prior to initiating antihypertensive therapy is very important for developing antihypertensive strategies and

**Table 1 Impact of frailty on prognosis of older hypertensive patients.**

| Author | Study design | Country | Sample size | Age (years) | Frailty assessment | Follow-up time | Frailty prevalence in hypertensive patients | The adverse effects of frailty on hypertensive patients |
|---|---|---|---|---|---|---|---|---|
| *Pajewski et al. (2016)* | Longitudinal | USA | 9,306 | ≥50 | 36-item frailty index | 3.26 years | 27.60% | Compared to fit adults, adjusted HR for self-reported falls, injurious falls and ACM were 1.72 (95% CI: [1.55–1.91]), 1.75 (95% CI: [1.35–2.28]) and 2.45 (95% CI: [2.20–2.74]) in frail adults, respectively. |
| *Bromfield et al. (2017)* | Longitudinal | USA | 5,236 | ≥65 | Low BMI, cognitive impairment, depressive symptoms, exhaustion, impaired mobility, history of falls | A median of 6.4 years | – | Compared to patients without frailty indicators, the adjusted HR for serious fall injury in patients with 1, 2 or ≥3 frailty indicators were 1.18 (95% CI: [0.99–1.40]), 1.49 (95% CI: [1.19–1.87]) and 2.04 (95% CI: [1.56–2.67]), respectively. |
| *Ma et al. (2018)* | Longitudinal | China | 1,111 | ≥60 | CGA-FI | 8 years | 19.60% | Frailty was related to a higher 8-year mortality, unadjusted HR = 3.40 (95% CI: [2.77–4.17]), and after adjusted for age and sex, HR = 2.61 (95% CI: [2.11–3.23]). |
| *Li et al. (2023)* | Longitudinal | USA | 2,177 | Mean age: 70.03 | Weakness, exhaustion, low physical activity, shrinkage, slowness | 3 years | 17.81% | Compared to non-frail hypertensive patients, the risk of ACM for frail patients aged ≥65 years and <65 years were 3.02 (95% CI: [2.50–3.65]) and 2.15 (95% CI: [1.43–3.25]), respectively. |

**Note:**
USA, the United States of America; HR, hazard ratio; CI, confidence interval; ACM, all-cause mortality; BMI, body mass index; CGA-FI, comprehensive geriatrics assessment-frailty index.

optimizing BP management, which could reduce adverse health outcomes and alleviate the burden of healthcare systems. Frailty screening and evaluation should be conducted for people aged ≥70 years or those who have experienced unintentional weight loss (≥5%) within the past year (*Morley et al., 2013*). In recent years, many international hypertension management guidelines or expert consensus have recommended to assess frailty status before initiating antihypertensive treatment in older people. Although many tools for assessing frailty have been developed worldwide, a gold standard is still lacking at present.

Currently, the Fried phenotype and Frailty Index (FI) are two most widely used tools for frailty assessment. The Fried phenotype originated in the United States and includes five indicators: slowness, weight loss, weakness, exhaustion, and low activity (*Fried et al., 2001*). Older people with three or more of the aforementioned indicators were defined as frailty, those with one or two components were defined as pre-frailty, and those without any components were defined as non-frailty. Subsequently, this method has been validated and widely used in many countries (*Ding, 2017*; *Joosten et al., 2014*). However, social psychology and cognitive function were not included in the Fried phenotype (*Bieniek,*

*Wilczynski & Szewieczek, 2016*), and slowness and weakness may not accurately reflect the actual status of patients with acute illness or some special conditions. The FI was developed by *Mitnitski, Mogilner & Rockwood (2001)*. FI is an age-related cumulative health deficit model, it evaluates frailty across many dimensions: physical, psychological, cognitive function, social function, and others (*Mitnitski, Mogilner & Rockwood, 2001*; *Rockwood & Mitnitski, 2007*). Each item of FI was assigned a score of 0–1, and an individual's FI was calculated as the total score of all items divided by the number of items included. The FI ranged from 0–1, and the greater the FI was, the more severity of the frailty status. Subsequently, researchers have appropriately modified the items of constructing FI according to different study populations, and variations exist in the definition of frailty according to FI cut-off values across different studies (*Searle et al., 2008*; *Gu et al., 2009*; *Shrauner et al., 2022*). *Liu et al. (2022)* used FI to assess frailty status and found FI was significantly associated with cardiovascular disease (CVD) in older adults, which emphasize the importance of frailty screening and effective interventions for improving primary prevention of CVD. The FI contains multiple health dimensions that can be used to evaluate the frailty status of patients relatively accurate. However, the inclusion of a substantial number of items may impede its practicality for evaluating frailty in busy clinical settings (*Aprahamian et al., 2017*).

In addition to the two commonly used frailty assessment tools, many others have been developed. Table 2 summarizes several widely used tools for assessing frailty (*Morley, Malmstrom & Miller, 2012*; *Bilotta et al., 2012*; *Gobbens et al., 2010*; *Bielderman et al., 2013*; *Ma et al., 2019*; *Satake et al., 2016*; *Pilotto et al., 2008*; *Daniels et al., 2012*; *Rolfson et al., 2006*). The ideal frailty screening and assessment tools should be selected according to the clinical context, and the characteristics of the subjects (*Richter et al., 2022*). Developing a sensitive and rapid tool to accurately identify frailty status among elderly people is very important in the future.

# HYPERTENSION MANAGEMENT IN FRAIL OLDER ADULTS

## Purpose of antihypertensive therapy

The purpose of antihypertensive therapy is to reduce hypertension-related damage to the heart, brain, kidney, fundus oculi, peripheral vessels and other target organs, improve the quality of life of patients and prolong their life. The primary focus of antihypertensive treatment in elderly patients is to achieve the target of SBP, as elevated SBP is a predominant characteristic of this population. BP should be closely monitored during treatment to prevent the occurrence of adverse events associated with hypotension and drugs.

## Benefits and risks of antihypertensive therapy

The benefits and risks of antihypertensive therapy in frail elderly patients with hypertension are controversial. Several studies have suggested that older people could benefit from aggressive or intensive antihypertensive therapy. Hypertension in the Very Elderly Trial (HYVET) enrolled 3,845 hypertensive patients aged ≥80 years with SBP > 160 mmHg, and the results demonstrated that active antihypertensive treatment (target

**Table 2 Tools for assessing frailty in older people.**

| Instrument | Country of origin | Components | Frailty classification criteria |
|---|---|---|---|
| FRAIL (*Morley, Malmstrom & Miller, 2012*) | USA | Five items: fatigue, resistance, ambulation, illness, and loss of weight | Frailty: ≥3 items; pre-frailty: 1–2 items; robust: 0 item |
| SOF (*Bilotta et al., 2012*) | USA | Three items: weight loss, self-reported poor energy, inability to rise from a chair five times without using the arms | Frailty: ≥2 items; pre-frail: 1 item; robust: 0 item |
| FSQ (*Ma et al., 2019*) | China | Four items: slowness, weakness, inactivity, exhaustion | Frailty: ≥3 items; pre-frailty: 1–2 items; robust: 0 item |
| KCL (*Satake et al., 2016*) | Japan | Seven dimensions (25 items): physical function, nutrition status, oral function, cognitive function, depressive mood, instrumental and social activities of daily living | Frailty: scores ≥8; pre-frail: scores range from 4 to 7; non-frail: scores ≤3 |
| GFI (*Bielderman et al., 2013*) | The Netherlands | Self-reported in four domains: physical, cognitive, social and psychological | Frailty: scores ≥4 |
| TFI (*Gobbens et al., 2010*) | The Netherlands | Self-reported in three domains: physical, psychological and social | Frailty: scores ≥5 |
| SPQ (*Daniels et al., 2012*) | Canada | Six self-reported: living alone, polypharmacy, mobility, eyesight, hearing, memory | Frailty: scores ≥2 |
| EFS (*Rolfson et al., 2006*) | Canada | Nine domains: cognition, general health status, functional independence, social support, medication use, nutrition, mood, continence, and functional performance | The score ranged from 0 to 17, and a higher score represents a higher degree of frailty. |
| MPI (*Pilotto et al., 2008*) | Italy | Comorbidity, nutrition status, mental status, number of medications, ADL, IADL, social support network, and the risk of developing pressure sores | Frailty: >0.66; pre-frailty: 0.33–0.66; robust: <0.33 |

**Note:**
FRAIL, fatigue, resistance, ambulation, illnesses, and loss of weight; USA, the United States of America; SOF, study of osteoporotic fracture; FSQ, frailty screening questionnaire; KCL, Kihon checklist; GFI, groningen frailty indicator; TFI, tilburg frailty index; SPQ, sherbrooke postal questionnaire; MPI, multidimensional prognostic index; ADL, activities of daily living; IADL, instrumental activities of daily living.

BP < 150/80 mmHg) reduced the incidence of fatal and non-fatal stroke by 30%; furthermore, the incidence of stroke-related death, death from any cause, cardiovascular-related death, and heart failure was reduced by 39%, 23%, 21%, and 64%, respectively (*Beckett et al., 2008*). However, HYVET only included older people with relatively good physical and cognitive status, and those requiring nursing care were excluded. The Systolic Blood Pressure Intervention Trial (SPRINT) showed that compared with standard antihypertensive treatment (SBP ≤ 140 mmHg), intensive antihypertensive treatment (SBP ≤ 120 mmHg) significantly reduced the risk of fatal and nonfatal cardiovascular events and death from any cause in hypertensive patients aged ≥75 years (*Williamson et al., 2016*). Notably, SPRINT excluded individuals with type 2 diabetes, a history of stroke, symptomatic heart failure within the past 6 months or ejection fraction <35%, dementia, expected survival <3 years, and weight loss >10% within the past 6 months. Another study from China also showed that intensive BP treatment (target SBP: 110–130 mmHg) significantly reduced the incidence of cardiovascular events compared with standard treatment (target SBP: 130–150 mmHg) in people aged 60–80 years (*Zhang & Cai, 2022*). *Corrao et al. (2017)* demonstrated that better adherence to antihypertensive therapy could significantly reduce the risk of cardiovascular events and mortality in older hypertensive patients aged >85 years. In addition, a meta-analysis revealed that

antihypertensive treatment could prevent 34% of strokes, 39% of heart failures and 22% of major cardiovascular events in patients aged ≥80 years (*Gueyffier et al., 1999*).

Noticeably, aggressive antihypertensive therapy has certain risks for frail elderly individuals as most of this population has multiple comorbidities (*Muller et al., 2014*), especially for those with terminal organ failure, severe functional dependency or dehydration. Several studies have shown that maintaining a slightly higher BP may be more beneficial for frail older patients. A study performed by *Sabayan et al. (2013)* demonstrated that a higher SBP, mean arterial pressure, and pulse pressure were associated with a lower risk of stroke among patients aged 85 years with impaired physical functioning. A prospective cohort study revealed that hypertension was not related to increased mortality in patients aged 75–84 years with moderate/severe frailty or >85 years; in addition, this study revealed that a BP < 130/80 mmHg was related to increased mortality in primary-care patients aged ≥75 years, regardless of the baseline frailty status (*Masoli et al., 2020*). A meta-analysis also demonstrated that in frail people aged >65 years, there was no significant difference in mortality between those with SBP < 140 mmHg and those with SBP > 140 mmHg (*Todd et al., 2019*).

In most cases, SBP increases with age in people aged ≥60 years, while DBP remains stable or even drops spontaneously (*Benetos, Petrovic & Strandberg, 2019*). Therefore, isolated systolic hypertension is the predominant form of hypertension in older people. Strict SBP control may result in a significant reduction in DBP, and increase the risk of inadequate perfusion to vital tissues and organs. In addition, a white-coat effect is frequently observed in elderly individuals, which may increase the risk of overtreatment (*Ishikawa et al., 2011*). Elderly hypertensive patients are not only at risk of hypertension-related adverse events, but also suffer from hypotension-related adverse events due to overtreatment, such as syncope, falls, electrolyte imbalances, acute kidney injury, and fractures (*Muller et al., 2014*; *Angelousi et al., 2014*; *Sink et al., 2018*), especially for frail older people (*Muller et al., 2014*). Therefore, frailty assessment is very important in determining antihypertensive treatment strategies for this population. Frail older hypertensive patients were often excluded from randomized controlled trials due to comorbidities or other causes (*Williams et al., 2018*). Moreover, current research primarily focused on when to start antihypertensive therapy, few studies explored when to reduce or stop antihypertensive drugs to avoid excessive BP reduction in frail elderly patients. More relevant studies are needed in the future to draw reliable conclusions.

## Initiating antihypertensive treatment and BP targets

Because of multiple comorbidities and physiological changes resulting from aging, the management of BP in frail elderly hypertensive patients differs from that in the general population. There is ongoing debate regarding the initiation of antihypertensive treatment and the target BP value. Multiple international guidelines and expert consensuses recommended assessing frailty status before initiating treatment in elderly patients, and then developing individualized antihypertensive treatment strategies according to the risk/benefit of older people. The consensus guidelines from Canada (*Mallery et al., 2014*) suggested that antihypertensive therapy should be initiated when SBP ≥ 160 mmHg, and

medications can be reduced when SBP < 140 mmHg if there are no special conditions. The 2017 American College of Cardiology guidelines (*Whelton et al., 2018*) recommended that for elderly patients aged >65 years and for those with a limited life expectancy, the intensity of antihypertensive therapy and choice of antihypertensive drugs should be based on clinical judgment and a comprehensive risk-benefit assessment. The 2018 European Society of Cardiology guidelines (*Williams et al., 2018*) recommended that for older patients aged ≥65 years, the target SBP is 130–139 mmHg, and DBP < 80 mmHg if the patient can tolerate. The 2020 International Society of Hypertension Global Hypertension Practice Guidelines (*Unger et al., 2020*) suggested that antihypertensive therapy should be initiated when BP ≥ 160/100 mmHg and that the BP target is <140/90 mmHg if tolerated in people aged ≥65 years. The 2023 European Society of Hypertension guidelines (*Mancia et al., 2023*) recommend that for frail patients, the initiation of drug treatment and the target BP should be individualized. According to the Chinese guidelines for the management of hypertension in elderly individuals, personalized antihypertensive treatment should be tailored for older patients with frailty, and the primary target of BP should be <150/90 mmHg (*Hypertension Branch of Chinese Geriatric Society, Beijing Association of Hypertension Prevention and Treatment, National Clinical Research Center for Geriatric Diseases, 2023*). Table 3 summarizes the recommendations from recent guidelines and consensuses for managing BP in frail older adults (*Williams et al., 2018*; *Mallery et al., 2014*; *Whelton et al., 2018*; *Unger et al., 2020*; *Mancia et al., 2023*; *Hypertension Branch of Chinese Geriatric Society, Beijing Association of Hypertension Prevention and Treatment, National Clinical Research Center for Geriatric Diseases, 2023*; *Visseren et al., 2021*).

## Antihypertensive drugs for older patients with frailty

The choice of antihypertensive drugs for frail older people is not significantly different from that for the general population. Thiazide diuretics, calcium channel blockers (CCBs), angiotensin converting enzyme inhibitors (ACEIs), or angiotensin receptor blockers (ARBs) could serve as initial or long-term maintenance pharmacotherapies (*Williams et al., 2018*; *Mancia et al., 2023*). Unless clinically indicated by comorbidities, beta blockers should not be considered first-line medications because they have adverse effects on cardiovascular disease outcomes in people aged ≥60 years (*Oliveros et al., 2020*). Priority should be given to specific classes of antihypertensive drugs according to the patient's risk factors and comorbidities (*Mancia et al., 2023*). Table 4 summarizes the optimal choice of antihypertensive drugs for elderly hypertensive patients in specific clinical situations.

Studies performed by *Di Bari et al. (2004)* and *Onder et al. (2002)* both demonstrated that ACEIs improved the muscle strength of older patients, suggesting ACEIs may be beneficial for frail older adults with a decline in physical function. *Li et al. (2010)* conducted a prospective cohort study in 819,491 American veterans aged ≥65 years and reported that ARBs significantly reduced the risk of Alzheimer's disease and dementia compared with other cardiovascular drugs. In addition, *Pozos-Lopez, Patricia Navarrete-Reyes & Alberto Avila-Funes (2011)* reported an inverse association between frailty status and ARBs use in women; however, this was a cross-sectional study with a

**Table 3 Guidelines/consensus for the management of hypertension in frail older people.**

| Organization | Year | Initiating antihypertensive treatment | Antihypertensive drugs | BP target | Evidence-based |
|---|---|---|---|---|---|
| Canada consensus guideline | 2014 | Frail older adults start antihypertensive treatment when SBP ≥ 160 mmHg | In general, use no more than two medications | A seated SBP: 140–160 mmHg, if the patient is severely frail and has a limited life expectancy, SBP could be controlled at 160–190 mmHg | Cohort studies/ guidelines/ RCTs |
| ACC/AHA | 2017 | Noninstitutionalized ambulatory community-living adults aged ≥65 years should start antihypertensive treatment when SBP ≥ 130 mmHg | Antihypertensive therapy with two agents should be cautiously, while closely monitoring for orthostatic hypotension and falls | Age ≥65 years with high burden of comorbidity, limited life expectancy: assess risk and benefit | RCTs/ cohort studies |
| ESC/ESH | 2018 | For people aged >80 years start antihypertensive therapy when SBP >160 mmHg | Monotherapy is recommended for initial therapy in frail older patients | For patients aged ≥65 years, SBP: 130–139 mmHg, and DBP < 80 mmHg if tolerated | Cohort studies/ RCTs |
| ISH | 2020 | SBP: 140–159 and/or DBP: 90–99 mmHg: start drug treatment for patients with high-risk or with CVD, CKD, DM or HMOD; SBP ≥ 160 and/or DBP ≥ 100 mmHg: drug treatment in all patients | Monotherapy is recommended for very old (>80 years) or frail patients | Aged ≥65 years: BP < 140/90 mmHg if tolerated | Cohort studies/ RCTs/ guidelines |
| ESC | 2021 | SBP: 140–159 and/or DBP:90–99 mmHg: drug treatment based on absolute CVD risk, estimated lifetime benefit, and the presence of HMOD; SBP ≥ 160 and/or DBP ≥ 100 mmHg: drug treatment | Monotherapy is recommended in initial therapy for frail patients or people aged >80 years | Age ≥70 years: SBP < 140 mmHg and down to 130 mmHg if tolerated, DBP < 80 mmHg | Cohort studies/ RCTs/ guidelines/ expert consensus |
| CSG/CSH | 2023 | Individualized BP management strategy was recommended | Monotherapy is recommended in initial therapy | The primary target is BP < 150/90 mmHg, but avoid <130 mmHg | Cohort studies/ RCTs/ guidelines/ expert consensus |
| ESH | 2023 | Patients with moderate functionality impairment and partial loss of autonomy: start treatment when SBP ≥ 160 mmHg; patients with severe loss of functionality/autonomy: indication of treatment should be individually decided according to symptoms, comorbidities and polypharmacy | Initiation with monotherapy was recommended in frailty patients | SBP: 140–150 mmHg for patients with moderate functionality impairment and partial loss of autonomy; considering progressive deprescription if SBP < 130 mmHg or occur orthostatic hypotension for patients with severe loss of functionality | Cohort studies/ RCTs/ guidelines/ expert consensus |

Note:
ACC, American College of CardIology; AHA, American Heart Association; ESC, European Society of Cardiology; ESH, European Society of Hypertension; ISH, International Society of Hypertension; CSG, Chinese Society of Geriatrics; CSH, Chinese Society of Hypertension; RCTs, randomized controlled trials; SBP, systolic blood pressure; BP, blood pressure; DBP, diastolic blood pressure; CVD, cardiovascular disease; CKD, chronic kidney disease; DM, diabetes mellitus; HMOD, hypertension-mediated organ damage.

relatively small sample size. A study from Taiwan (*Chuang et al., 2016*) showed that CCBs significantly reduced the risk of frailty and pre-frailty in elderly patients, suggesting that CCBs may have a protective effect on frail elderly hypertensive patients. However, this was a cross-sectional study with a relatively small sample size, thus more research is needed to prove the reliability of the findings in the future. Diabetes, cardiovascular disease and

**Table 4 The optimal choice of antihypertensive drugs for elderly hypertensive patients in specific clinical situations.**

| Specific clinical situations | Antihypertensive drugs |
|---|---|
| Asymptomatic target organs damage | |
|     LVH | CCBs/ACEIs/ARB/ARNIs |
|     Atherosclerosis | CCBs/ACEIs/ARBs |
|     Mild renal insufficiency | ACEIs/ARBs/ARNIs |
|     Microalbuminuria | ACEIs/ARBs |
| Clinical cardiovascular events | |
|     Old myocardial infarction | ACEIs/ARBs/beta-blockers |
|     Angina | CCBs/beta-blockers |
|     Heart failure | ACEIs/ARBs/diuretics/MRAs/beta-blockers/ARNIs |
|     Atrial fibrillation, ventricular rate control | Non-dihydropyridine CCBs/beta-blockers |
|     Atrial fibrillation, prevention | ACEIs/ARBs/MRAs/beta-blockers |
|     Aortic aneurysm | Beta-blockers/ARBs |
|     Peripheral arterial disease | CCBs/ACEIs/ARBs |
|     Renal injury/proteinuria | ACEIs/ARBs/ARNIs |
| Others | |
|     Isolated systolic hypertension | CCBs/diuretics |
|     Diabetes | ACEIs/ARBs |
|     Metabolic syndrome | CCBs/ACEIs/ARBs |
|     Prostatic hyperplasia | α-receptor inhibitors |

**Notes:**
LVH, left ventricular hypertrophy; CCBs, calcium channel blockers; ACEIs, angiotensin-converting-enzyme inhibitors; ARB, angiotensin receptor blockers; ARNI, angiotensin receptor-neprilysin inhibitors; MRA, mineralocorticoid receptor antagonists.

chronic kidney disease are prevailing comorbidities in frail patients, sodium-glucose cotransporter 2 (SGLT2) inhibitors have hypoglycemic effect, cardiac and renal protective effects. The efficacy and safety of SGL T2 inhibitors in frail elderly subjects has been confirmed in recent years (*Abdelhafiz & Sinclair, 2020*; *Sasaki, 2019*; *Villarreal et al., 2023*). SGLT2 inhibitors may be considered novel anti-frailty drugs (*Santulli et al., 2023*).

Frailty has a negative impact on adherence to antihypertensive therapy in elderly patients (*Chudiak, Jankowska-Polanska & Uchmanowicz, 2017*; *Pobrotyn et al., 2021*; *Jankowska-Polanska et al., 2018*). On the one hand, frail elderly hypertensive patients often have other chronic diseases, increasing the risks of polypharmacy and drug interactions (*Kang et al., 2017*; *Gnjidic et al., 2012*); on the other hand, due to the decline in physical function and changes in pharmacokinetics, frail elderly patients have a reduced tolerance to drugs and a significantly increased risk of adverse drug reactions (*McLean & Le Couteur, 2004*; *Hilmer, McLachlan & Le Couteur, 2007*). Poor adherence to antihypertensive treatment may increase the risk of hypertension-related adverse health outcomes (*Burnier et al., 2013*; *Corrao et al., 2011*). Therefore, it is important to optimize medications and reduce unnecessary drugs in frail older people. The drug treatment of elderly patients with hypertension should adhere to the following principles (*Williams et al., 2018*; *Whelton et al., 2018*; *Unger et al., 2020*; *Mancia et al., 2023*; *Hypertension Branch of Chinese*

*Geriatric Society, Beijing Association of Hypertension Prevention and Treatment, National Clinical Research Center for Geriatric Diseases, 2023*; *Visseren et al., 2021*): (1) initial treatment usually starts with a small effective dose, and the dosage can be gradually increased if BP is not well controlled; (2) long-acting antihypertensive drugs, which can effectively manage BP both day and night, should be recommended; (3) single-tablet compound preparations may enhance patients adherence, and combination therapy with two or more drugs can be employed if the efficacy of monotherapy is unsatisfactory; and (4) a personalized antihypertensive strategies should be developed according to frailty status, comorbidities, response to drugs, tolerance, and risk/benefit of older people.

Frail patients generally do not meet the inclusion criteria for randomized controlled trials (RCTs), limiting the generalizability of RCT findings and making it difficult to estimate the safety and efficacy of antihypertensive drugs for this special population (*Benetos, Petrovic & Strandberg, 2019*). Therefore, well-designed clinical trials involving frail older people are needed in the future to provide robust antihypertensive strategies for this population.

## MANAGEMENT OF FRAILTY IN OLDER HYPERTENSIVE PATIENTS

Frailty changes dynamically over time and it is partially reversible (*Kojima et al., 2019*; *Hoogendijk et al., 2019*). Given that frailty has an important influence on the management of hypertension in elderly patients, it is necessary to identify and assess frailty at an early stage and take appropriate measures to prevent it. Genes, aging, gender, lifestyle, nutrition, chronic diseases, psychosocial support, sociodemographic, and polypharmacy are associated with the occurrence and development of frailty (*Dent et al., 2019*; *He et al., 2019*; *Zazzara et al., 2019*; *Angulo et al., 2020*; *Lochlainn et al., 2021*).

Effective management of frailty could reduce the burden on health systems and families (*Dent et al., 2019*). Frailty can be managed in the following ways in elderly hypertensive patients (*Dent et al., 2019*; *Clegg et al., 2013*; *Angulo et al., 2020*; *Lochlainn et al., 2021*; *Dent et al., 2017*): (1) physical activity is widely regarded as the optimal approach for preventing and managing frailty, older people with frailty should appropriately participate in aerobic exercise, resistance training, stretching exercise, and balance training; (2) focusing on the management of chronic diseases and polypharmacy. The effective management of multiple chronic diseases necessitates collaboration among geriatrics, cardiologists, endocrinologists, primary care providers, and doctors in other departments; in addition, polypharmacy should be addressed by reducing or deprescribing any inappropriate or unnecessary medications; (3) supplementation with protein/calories should be considered when frail older adults experience weight loss or are diagnosed with malnutrition; (4). vitamin D supplementation should be considered for older people with vitamin D deficiency; (5) psychosocial support should be provided for the elderly population. In addition, hyperglycemia drives the transition from pre-frailty to frailty in hypertensive patients (*Mone et al., 2023*), and it drives physical impairment in frail hypertensive (*Pansini et al., 2022*), therefore, strict blood glucose monitoring and management is very important in preventing frailty.

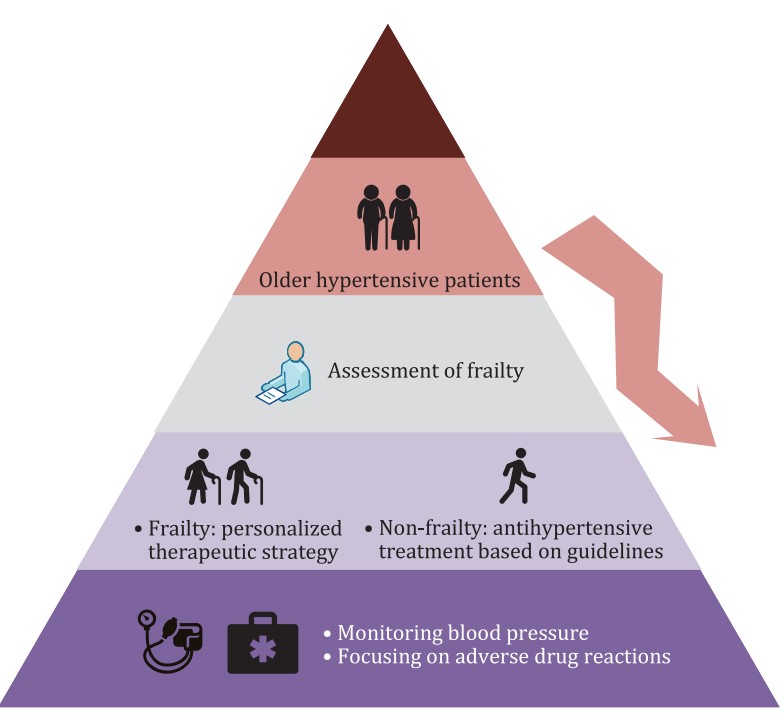

**Figure 1 Blood pressure management process in older people with hypertension.**

## CONCLUSION

Frailty assessment is necessary for elderly patients with hypertension. For frail older hypertensive patients, a personalized antihypertensive therapeutic strategy should be developed according to the patient's condition, and BP and drug-related adverse events should be closely monitored (Fig. 1). Although great progress has been made in the fields of frailty and hypertension in recent decades, there remains a dearth of international guidelines or consensuses regarding BP management in older people with frailty. In the future, more longitudinal studies based on frail older people are needed to provide strong evidence of antihypertensive therapeutic strategies for this population, and the potential causal relationship between frailty and hypertension requires further investigation.

### Funding

The study was supported by the project from Sichuan Science and Technology Program, China (Grant No. 2022YFS0186) and the National Natural Science Foundation of China (Grant No. 81600299). The funders had no role in study design, data collection and analysis, decision to publish, or preparation of the manuscript.

### Grant Disclosures

The following grant information was disclosed by the authors:
Sichuan Science and Technology Program, China: 2022YFS0186.
National Natural Science Foundation of China: 81600299.

## Competing Interests

The authors declare that they have no competing interests.

## Author Contributions

- Liying Li conceived and designed the experiments, prepared figures and/or tables, authored or reviewed drafts of the article, and approved the final draft.
- Linjia Duan conceived and designed the experiments, performed the experiments, prepared figures and/or tables, authored or reviewed drafts of the article, and approved the final draft.
- Ying Xu conceived and designed the experiments, prepared figures and/or tables, authored or reviewed drafts of the article, and approved the final draft.
- Haiyan Ruan performed the experiments, prepared figures and/or tables, and approved the final draft.
- Muxin Zhang performed the experiments, prepared figures and/or tables, and approved the final draft.
- Yi Zheng analyzed the data, authored or reviewed drafts of the article, and approved the final draft.
- Sen He conceived and designed the experiments, authored or reviewed drafts of the article, and approved the final draft.

## Data Availability

This article is a literature review.

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
