# Peer review of "Hypertension in frail older adults: current perspectives"

_PeerJ, doi:10.7717/peerj.17760_

## Round 0.1 · original submission · Major Revisions

Please address all the comments from the reviewers and resubmit the revised manuscript.

**Language Note:** The review process has identified that the English language must be improved. PeerJ can provide language editing services - please contact us at [email protected] for pricing (be sure to provide your manuscript number and title). Alternatively, you should make your own arrangements to improve the language quality and provide details in your response letter. – PeerJ Staff

·

Basic reporting

This is a very well written and informative review by Li et al, illustrating different parameters and risk management of frailty and their association with hypertension in elderly. Overall, this is very compact and informative, however, I would suggest a few additional things.

For the better understanding of the larger audience please include a table mentioning all the available treatments/medications for hypertension. These treatment/medication list should comprise both frail and non-frail individuals with hypertension.

Please include a graphical abstract or a schematic scheme to highlight the main aim/objective and conclusion of this manuscript.

In addition, the manuscript should be assessed by language editing service to improve the quality of English. Many grammatical errors were observed throughout the manuscript.

Experimental design

No comments.

Validity of the findings

No comments.

Additional comments

No further comments.

Reviewer 2 ·

Basic reporting

The presentation and critical interpretation of results of previous studies should be improved.

Many recent reports dealing with the topic of the review have not been mentioned.

Experimental design

The Authors should incorporate a pictorial or cartoon representation of the topics discussed in the Review in order to facilitate the comprehension and increase the overall impact of the manuscript.

Validity of the findings

Overall, the topic of this review is of relevance for the scientific community and I think worth being published. However, the manuscript in its current form appears rather preliminary and not really carefully crafted, resembling more a "draft" than a final version.

Additional comments

English language (syntax, grammar, correct choice of words, correct use of adjectives and adverbs) needs significant editing throughout the text. Professional assistance must be sought.
"A person with three or more of the five components will be defined frailty".



The following pertinent topics should be mentioned/discussed:

- Management of Hypertension in the Elderly and Frail Patient.
- Antihypertensive Polypharmacy in Frailty
- Physical decline and cognitive impairment in frail hypertensive elders during COVID-19
- Relationship between diet quality scores and the risk of frailty and mortality
- Functional and clinical importance of SGLT2-inhibitors in frailty
- Association between Frailty and Cognitive Impairment in Patients with Hypertension
- Hyperglycemia and physical impairment in frail hypertensive older adults.
- Insulin resistance drives cognitive impairment in hypertensive prediabetic frail elders
- Frailty Modifies the Association of Hypertension with Cognition
- Global cognitive function correlates with P-wave dispersion in frail hypertensive older adults.
- Hyperglycemia drives the transition from pre-frailty to frailty in older adults
- Arginine improves cognitive impairment in hypertensive frail patients
- Frailty Index and Cardiovascular Disease among Middle-Aged and Older Adults

Reviewer 3 ·

Basic reporting

The manuscript covers a wide range of topics relevant to hypertension in frail older adults, including management strategies, drug choices, and risk factors. However, I have some questions-

Clarity and Structure: Some sections have long and complex sentences that could be simplified for better readability and understanding.
Consistency in Terminology: Ensuring consistent use of terms throughout the manuscript enhances clarity.

Experimental design

Comprehensive Literature Search: The authors describe a thorough search across multiple databases, which is fundamental for a narrative review.
Areas for Improvement:
Selection Criteria and Bias: The authors should clearly outline their criteria for including and excluding studies in the review. This helps in assessing potential biases in the selection of literature.
Diversity of Sources: It's important to ensure a diverse range of sources, including studies with conflicting findings, to provide a balanced overview.

Validity of the findings

Correlation vs. Causation: If the review discusses relationships between frailty and hypertension, it's important to distinguish between correlation and causation, especially since this can be a complex issue in geriatric populations.
Discussion of Study Limitations: The authors should discuss the limitations of the studies they cite, especially regarding how these limitations might affect the review's conclusions.

Additional comments

Relevance to Current Practices: The manuscript is timely and relevant, considering the aging global population and the complexities of managing chronic conditions in the elderly.
Practical Implications: It would be beneficial if the manuscript could provide more practical guidance or recommendations for clinicians based on the reviewed literature.
Future Research Directions: Suggestions for future research, particularly in areas where there is currently a lack of consensus or limited data, would be valuable.
Multidisciplinary Approach: Given the complexity of managing frail elderly patients, a discussion on the role of a multidisciplinary approach, including geriatricians, cardiologists, primary care providers, and others, could add value.

Reviewer 4 ·

Basic reporting

The review , Hypertension in the frail older adults: Current perspectives" is a comprehensive review with a thorough literature search. This topic has not been very recently reviewed ( Last reviewed in 2020: Hypertension in older adults: Assessment, management, and challenges). This review dives deep into the relationship between Hypertension and frailty and the management of hypertension in older adults with frailty. The organization of the sections in the review makes the review easy to read.

However, the grammar and English needs improvement throughout the review to increase the readability of the review.
Eg: Few Minor Comments:

1) The title would be better by removing the "the" . " Hypertension in Frail Older Adults: Current perspective"

2) Line 77 and 78: Use either percentage or fractions for ease of reading.

3) Line 111; 115-116; 251: Sentence construction needs improvement.

Experimental design

The survey method is comprehensive and the sources and recent publications are adequately cited. The review is organized well into relevant sections that make sense. However, the authors missed a few recent citations that should be considered to be included and elaborated in the review.

Oliveros E, Patel H, Kyung S, Fugar S, Goldberg A, Madan N, Williams KA. Hypertension in older adults: Assessment, management, and challenges. Clin Cardiol. 2020 Feb;43(2):99-107. doi: 10.1002/clc.23303. Epub 2019 Dec 11. PMID: 31825114; PMCID: PMC7021657.

Benetos A, Petrovic M, Strandberg T. Hypertension Management in Older and Frail Older Patients. Circ Res. 2019 Mar 29;124(7):1045-1060. doi: 10.1161/CIRCRESAHA.118.313236. PMID: 30920928.

Validity of the findings

The conclusions of the review are adequate. With the increase in the longevity of the population, an international guideline describing BP management in old people with frailty is important and the review mentions this key point. It also calls for additional longitudinal studies to assess association of frailty and hypertension.

Additional comments

No additional comments

---

## Round 0.2 · Minor Revisions

Please respond to the remaining minor comments from the reviewers and re-submit the revised manuscript. Thanks

Reviewer 2 ·

Basic reporting

The Review is still missing a proper discussion of relevant topics.
The Authors ignored most of the pertinent reports that had been suggested by the Reviewers.

Experimental design

.

Validity of the findings

.

Additional comments

.

Reviewer 3 ·

Basic reporting

The authors have replied satisfactorily to my comments.

Experimental design

The authors have replied satisfactorily to my comments.

Validity of the findings

The authors have replied satisfactorily to my comments.

Additional comments

N/A

Reviewer 4 ·

Basic reporting

This version of the review manuscript is more readable and clear. The English and grammar has greatly improved, making the flow of the article better. Additional relevant references and information have been added. The authors have addressed all my previous comments.

However, there are some minor changes required.

The formatting of the tables, particularly table 1 and 3 will need improvement as some of the text is very close to the borders. Maybe adding them vertical (landscape mode) will help.

There is an extra line for Figure legend for figure one at line 729-730 (on the pdf version) that seems out of place and needs to be deleted.

The commonly used Acronym for Angiotensin-Converting Enzymes Inhibitors is ACEIs (capitalized I). Since this acronym is used widely in the articles and tables, please change it to ACEIs.

Experimental design

No comment

Validity of the findings

No Comment

Additional comments

No Comments

---

## Round 0.3 · accepted · Accept

Thank you for addressing all the reviewers' comments and resubmitting the revised manuscript.